# Preparation and Mechanism Analysis of High-Performance Humidity Sensor Based on Eu-Doped TiO_2_

**DOI:** 10.3390/s24134142

**Published:** 2024-06-26

**Authors:** Ling Zhang, Chu Chen, Hongyan Zhang

**Affiliations:** 1School of Physical Science and Technology, Xinjiang University, Urumqi 830017, China; zhy@xju.edu.cn (L.Z.); chenchu@xju.edu.cn (C.C.); 2Xinjiang Key Laboratory of Solid State Physics and Devices, Xinjiang University, Urumqi 830017, China

**Keywords:** humidity sensor, TiO_2_, doped Eu atom, density functional theory

## Abstract

TiO_2_ is a typical semiconductor material, and it has attracted much attention in the field of humidity sensors. Doping is an efficient way to enhance the humidity response of TiO_2_. Eu-doped TiO_2_ material was investigated in both theoretical simulations and experiments. In a simulation based on density functional theory, a doped Eu atom can increase the performance of humidity sensors by producing more oxygen vacancies than undoped TiO_2_. In these experiments, Eu-doped TiO_2_ nanorods were prepared by hydrothermal synthesis, and the results also confirm the theoretical prediction. When the doping mole ratio is 5 mol%, the response of the humidity sensor reaches 23,997.0, the wet hysteresis is 2.3% and the response/recovery time is 3/13.1 s. This study not only improves the basis for preparation of high-performance TiO_2_ humidity sensors, but also fills the research gap on rare earth Eu-doped TiO_2_ as a humidity-sensitive material.

## 1. Introduction

Humidity sensors play an important role in industry, agriculture, environmental monitoring, food safety, precision instrument storage and health monitoring [1,2,3]. To meet the needs in those fields, it is urgent that humidity sensors with fast response, easy manufacture, good stability and reliability are prepared [4]. There are many types of humidity sensors, such as resistance, capacitance, impedance and voltage (electrochemical, ion gradient) sensors [2,3,5,6]. Among these sensor types, impedance humidity sensors have high precision, good heat dissipation and large nonlinearity, so they can be widely used in automatic testing and control technology. For humidity sensors, sensitive material directly interacts with target molecules and has an important influence on performance. Compared to metal oxide semiconductors (MOSs), many humidity-sensing materials, such as halloysite nanotubes, attapulgite, sepiolite nanofibers, zirconium phosphate nanoplates and MXenes, also exhibit excellent humidity-sensing performance. But, in fact, MOSs have been widely used in humidity-sensitive material due to their adjustable bandgap, large specific surface area, abundant surface defects and reasonable charge transfer ability [5]. Furthermore, MOS-based humidity sensors show notable features such as small size, easy synthesis and low cost compared with other types of humidity sensors [7].

As a typical MOS, TiO_2_ has attracted much attention in humidity sensors due to its good corrosion resistance, excellent charge transfer ability, high thermal stability and easy synthesis [8,9,10]. In addition, the surface of TiO_2_ material is rich in oxygen vacancies and hydrophilic groups, which allows more water molecules to be adsorbed on the surface of TiO_2_ to accelerate their ionization, thus enhancing the response speed of the sensor [11,12,13]. The increase in the conductance of TiO_2_ by increasing the RH can be described as a sequence of dissociation and reduction. The reaction constitutes the homolytic dissociation of H_2_O and the reaction of the neutral H atom with the lattice oxygen. In addition to ionizing the lattice, the reaction releases the electron as a donor to the rooted OH group [14]. Therefore, it is interesting to improve the performance of TiO_2_ by increasing surface hydrophilic groups and oxygen vacancies.

Recently, some methods, including doping, recombination and modification, have been applied to control the structure and morphology of TiO_2_ to achieve better humidity-sensing performance [15,16]. Rare earth (RE) elements, such as europium (Eu), cerium (Ce), Erbium (Er) and dysprosium (Dy) [17,18,19], can easily adjust the magnetic, electrical and optical properties of TiO_2_ because RE ions in the 4F orbital are shielded from external 6S, 5P and 5D orbitals [20]. Therefore, RE ions are expected to improve the performance of resistive humidity sensors based on MOSs. As it is one of the most active RE metal elements, Eu doping makes it easier to modulate semiconductor bandgap width [21,22]. Furthermore, Eu ion doping may contribute to locating some Eu-rich compositions on the surface of TiO_2_ and further increase surface oxygen vacancies and hydrophilic groups of TiO_2_, which can absorb more water molecules to enhance the response of TiO_2_ humidity sensors [23]. In addition, the strong electric field around oxygen vacancies can accelerate the dissociation of water molecules so that more protons can migrate freely to improve the response speed of the sensor [24]. There is little information about the application of Eu-doped TiO_2_ in the design of resistive humidity sensors, and the relevant humidity-sensing mechanism is still unclear. Therefore, it remains a great challenge to control Eu-doped TiO_2_ nanostructures in an effective way to improve humidity-sensing performance.

In this paper, the study of density functional theory (DFT) is used to investigate the feasibility of TiO_2_ doped by Eu. The calculations show that the introduction of Eu ions into TiO_2_ can result in the formation of more oxygen vacancies on the surface of TiO_2_. Furthermore, the oxygen atoms around Eu atoms can attract hydrogen ions in the environment to form a hydroxyl, which is conducive to the capture of water molecules. In these experiments, Eu-doped TiO_2_ resistive humidity sensors with different mole ratios were prepared by hydrothermal synthesis to achieve better humidity-sensing performance. Hydrothermal synthesis is relatively simple, does not require high-temperature treatment and can obtain a complete crystal shape, uniform particle size and good dispersion, thus relatively reducing energy consumption. In addition, the required raw materials are relatively cheap and easy to obtain, and the resulting products have a uniform phase, high purity, good crystallization, high yield and controllable product morphology and size. A Eu-doped TiO_2_ humidity sensor has a response of 23,997.0 in the range of 0% to 95% RH, a humidity hysteresis loop of 2.3% and a response/recovery time of 3/13.1 s. This humidity sensor also shows high long-term stability and good anti-interference ability. In addition, the mechanism of proton migration on the Eu-doped TiO_2_ surface was also investigated by AC impedance spectroscopy.

## 2. Experimental Details

### 2.1. Materials and Characterizations

Titanium tetrachloride (TiCl_4_) and europium oxide (Eu_2_O_3_) were provided by Aladdin Reagent (Shanghai) Co., Ltd. Citric acid (C_6_H_8_O_7_) was purchased from Tianjin Zhiyuan Chemical Reagent Co., Ltd. The reagents in the experiment were all analytical grade (AR) and did not require further purification for use.

The microstructure of the material was obtained with an S-4800 scanning electron microscope (SEM, Hitachi, Japan). Structure analysis was performed using D8 Advance phase X-ray diffraction (XRD, Bruker, Germany). The elemental composition of the samples was analyzed on a Thermo ESCALAB 250Xi X-ray photoelectron spectrometer (XPS, Thermo Fisher Scientific, USA). Fourier transform infrared spectra of samples were measured with an Invenio R Fourier transform infrared spectrometer (FTIR, Bruker, Germany).

### 2.2. Preparation of Eu-TiO_2_

Different amounts of europium oxide (0.0352 g, 0.106 g, 0.176 g or 0.352 g) were separately added into 10 mL of hydrochloric acid, then stirred in a constant-temperature water bath at 90 °C for 30 min. This solution was marked as solution A. A total of 2 mL of titanium tetrachloride was added into 30 mL of deionized water, then stirred for 20 min. Then, 0.2 g of citric acid was added into the above titanium tetrachloride solution and stirred for 30 min. This solution was marked as solution B.

Cooled down to 25 °C, solution A was added into solution B and stirred for 30 min. The above mixed solution was transferred to an autoclave lined with 100 mL of polytetrafluoroethylene and placed in a constant-temperature oven at 180 °C for 6 h. Then, the mixture was cooled down to room temperature for 2 h, before the precipitate was centrifuged at 8000 rpm with DI and anhydrous ethanol, respectively, for 10 min, which was repeated three times. Finally, the precipitate was dried in a constant-temperature drying oven at 80 °C for 6 h to obtain Eu-TiO_2_ powder.

The molar ratios of europium oxide and titanium tetrachloride were 1 mol%, 3 mol%, 5 mol% and 10 mol%, named as 1-mol%-Eu-TO, 3-mol%-Eu-TO, 5-mol%-Eu-TO and 10-mol%-Eu-TO. TiO_2_ powder can be synthesized without adding solution A in the above synthesis process.

### 2.3. Humidity Sensor Tests

Figure 1a shows the schematic flow diagram of the preparation of Eu-doped TiO_2_. A total of 5 mg of Eu-TiO_2_ and 5 mL of deionized water were mixed and ground for 10 min to form a paste. Then, the paste was applied on the surface of Ag-Pd interdigital electrodes using a small brush. Then, the coated Ag-Pd IDEs were put into a drying oven and dried for 1 h at a temperature of 60 °C to obtain a Eu-TiO_2_ humidity sensor.

The schematic diagram of the Eu-TiO_2_ humidity-sensing test system can be found in Figure 1b. All characteristic parameters of the humidity sensor were measured with the Zennium X Electrochemical Workstation. The driving work voltage was set to be AC 1 V, and the frequency was 40 Hz to 100 kHz. During the test, different humidity environments, including 33%, 43%, 59%, 75%, 85% and 95% RH humidity environments, were provided by supersaturated solutions of MgCl_2_, K_2_CO_3_, NaBr, NaCl, KCl and KNO_3_, and a 0% RH environment was provided by a P_2_O_5_ desiccant. The humidity test elements were kept in an environment of 0–95% RH to complete the performance test of the humidity sensor, and indoor temperature was kept at 25 °C throughout the whole test. Furthermore, the sensor’s equilibrium time at each humidity level was approximately 100 s.

## 3. Results and Discussion

Figure 2a shows the XRD patterns of TiO_2_, 1-mol%-Eu-TO, 3-mol%-Eu-TO, 5-mol%-Eu-TO and 10-mol%-Eu-TO. All materials exhibit 11 characteristic diffraction peaks of rutile-phase TiO_2_ (JCPDS card no. 21-1276). It can be seen from the inset that the diffraction peaks of the (110), (101) and (211) crystal planes of Eu-TiO_2_ are shifted to a smaller angle direction compared to TiO_2_. With the increase in Eu content, the shifted angle of the diffraction peaks increases gradually. This phenomenon is due to the ionic radius of Eu being much larger than that of Ti, resulting in the expansion of the TiO_2_ lattice [25]. This shift also indicates that Eu is successfully doped into TiO_2_.

In Figure 2b, functional groups on the surface of TiO_2_ and Eu-TiO_2_ are measured and analyzed by FTIR. It can be observed that absorption peaks at 517 cm^−1^ of all samples are caused by tensile vibration of Ti–O–Ti. The absorption peak at 1615 cm^−1^ is due to the bending vibration of water molecules, and the broad absorption band at 3389 cm^−1^ is attributed to the stretching vibrational mode of –OH [26]. As shown in the inset of Figure 2b, the introduction of Eu makes the absorption peak of the Eu-TO samples at 3389 cm^−1^ gradually broaden, and the intensity of the absorption peak increases. It can be found that the absorption peak of 5-mol%-Eu-TO is the broadest, and the intensity of the absorption peak is the highest, indicating that the content of -OH is the highest. -OH is a hydrophilic functional group, which means that 5-mol%-Eu-TO has a strong ability to adsorb more water molecules and enhance the response to a certain extent.

The morphologies of TiO_2_ and 5-mol%-Eu-TO were tested and analyzed by scanning electron microscopy (SEM). In Figure 3a, TiO_2_ exhibits a rod-like structure with a length of about 240 nm, and the shape of the nanorods is regular. The 5-mol%-Eu-TO exhibits nanorods with a length of about 80–180 nm, and the nanorods are regular in shape and uniform in distribution, as shown in Figure 3b. Compared to TiO_2_, the smaller size of 5-mol%-Eu-TO means more active sites. The chemical composition of 5-mol%-Eu-TO was analyzed by EDS spectroscopy, as shown in Figure 3c. It can be observed that 5-mol%-Eu-TO is composed of Ti, O and Eu, and the mass percentages of each element are 40.63, 59.00 and 0.34, respectively. Figure 3d shows the mapping pattern of 5-mol%-Eu-TO, in which Ti, O and Eu elements are all presented, and the three elements are distributed uniformly, which means Eu-doped TiO_2_ was successfully prepared.

To further analyze the chemical states of the materials, the XPS spectra of Ti 2p and O 1s of TiO_2_ and 5-mol%-Eu-TO were tested, as shown in Figure 4. In Figure 4a, the peaks with binding energies of 457.70 eV and 463.41 eV belong to Ti 2p_3/2_ and Ti 2p_1/2_. The difference between binding energies of these two peaks is 5.71 eV, which indicates that Ti ions exist in the form of Ti^4+^ [27]. Compared to TiO_2_, the Ti 2p peak in 5-mol%-Eu-TO shifts in the lower binding energy direction, indicating that Ti-Ti bonds are formed in the samples. Figure 4b,c are XPS spectra of O 1s of TiO_2_ and 5-mol%-Eu-TO which were recorded to analyze different forms of oxygen. O 1s is decomposed into three peaks with binding energies of 528.78 eV, 529.20 eV and 530.78 eV, which represent lattice oxygen (Lo), oxygen vacancies (Vo) and chemisorbed oxygen (Co) [28,29], respectively. The comparison of the proportions of Lo, Vo and Co shows that the content of oxygen vacancies in 5-mol%-Eu-TO is much higher than that in TiO_2_, indicating that doping of Eu is beneficial in increasing the content of oxygen vacancies. It is well known that oxygen vacancies on the surface of materials can increase the adsorption capacity and decomposition rate of water molecules to enhance the response and response speed of humidity sensors. Therefore, considering the theoretical and the experimental results mentioned above, 5-mol%-Eu-TO is predicted to have better humidity-sensing performance in all samples.

Figure 5a shows the response of TiO_2_ and Eu-TiO_2_ humidity sensors in the range of 0% ~95% RH, and the operating frequency is 100 Hz. It can be observed that the impedance of the TiO_2_ humidity sensor decreases from 1,778,000 to 1230 Ω, and the impedance changes by three orders of magnitude. The introduction of Eu increases the impedance of the material as much as nearly tenfold at low humidity, which is mainly because doping of Eu increases the amount of -OH and number of oxygen vacancies on the surface of TiO_2_, which enhances the adsorption capacity of TiO_2_ for water molecules in low-humidity environments. The response (R) of the sensor can be calculated as R = (Z_0_ − Z_95_)/Z_95_, where Zx is the impedance value of the sensor at x% RH [30]. The responses of the TiO_2_, 1-mol%-Eu-TO, 3-mol%-Eu-TO, 5-mol%-Eu-TO and 10-mol%-Eu-TO humidity sensors are 1444.5, 2890.2, 6159.4, 23,997.0 and 5087.2, respectively. The response of the 5-mol%-Eu-TO humidity sensor is the highest, which is mainly because the doping of Eu promotes the generation of more oxygen vacancies, -OH and active sites on surface of the material. Hydroxyls and oxygen vacancies can allow more water molecules to be adsorbed on the surface to improve the response of the 5-mol%-Eu-TO humidity sensor. A strong electrostatic field exists near the oxygen vacancies, which can rapidly dissociate more water molecules, thereby increasing the conductivity of the sensor. In addition, the smaller particle size of 5-mol%-Eu-TO makes its specific surface area larger than that of TiO_2_, which improves the adsorption rate of the water molecules. All the above factors make the 5-mol%-Eu-TO humidity sensor exhibit the best response and linearity. The responses of the 1-mol%-Eu-TO, 3-mol%-Eu-TO and 10-mol%-Eu-TO humidity sensor are lower, which may be because it is not easy to induce more oxygen vacancies and -OH on the TiO_2_ surface with other amounts of Eu.

Working frequency is also an important indicator to measure the performance of a sensor. Figure 5b shows the response changes of 5-mol%-Eu-TO humidity sensor at a working frequency of 40 to 100 kHz. The frequency and impedance values of the sensor at different working frequencies are usually monotonic. At low frequencies (40 Hz), the response of the humidity sensor reaches the highest level, but the linearity of the sensor is relatively poor due to the inherent impedance of the material. At high frequencies (1k, 10k and 100k Hz), when humidity is lower than 43% RH, the humidity sensor has a poor response to water molecules, and only has a small response to water molecules in the range of 43% to 95% RH. Figure 5b shows that higher frequencies reduce the sensor’s performance, which is mainly because the polarization direction of water molecules is difficult to keep consistent with the changing direction of the high-frequency electric field. Meanwhile, it is difficult for the polarization speed of water molecules to keep up with changes in the electrical field direction at a high working frequency, resulting in a decrease in the dielectric constant and a decrease in its dependence on RH [30]. When the working frequency is 100 Hz, the response and linearity of the 5-mol%-Eu-TO sensor reach the best levels. Therefore, 100 Hz can be used as the ideal working frequency of this sensor, and subsequent tests were completed at this frequency.

Figure 5c shows the hysteresis characteristic curve of the 5-mol%-Eu-TO humidity sensor, which is also an important parameter to measure the reversibility of a sensor. The sensor has a strong ability to desorb water molecules in a high-humidity environment, and the hysteresis is controlled within a small range. The 5-mol%-Eu-TO sensor produces large humidity hysteresis in a dry environment (0% RH), which is affected by its morphology, so that the water molecules cannot be desorbed on the surface in time. The humidity hysteresis error can be calculated according to γH = Hmax/2FFS, where Hmax is the maximum difference of impedance, and FFS is the full-scale output. From this, the maximum hysteresis error of the 5 mol% Eu-TO humidity sensor at 0% RH can be calculated to be 2.3%.

The response and recovery speed of the sensor to humidity changes are also important for the performance of the humidity sensor. Figure 6a shows the response and recovery process of the 5-mol%-Eu-TO humidity sensor in the range of 0%~95% RH. The changes in curves in four consecutive adsorption and desorption processes are almost the same, which indicates that the sensor has good repeatability. Response or recovery time is defined as the time required for a 90% change in impedance value during adsorption or desorption. The sensor’s response time is 3 s and the recovery time is 13.1 s, as shown in Figure 6b. Compared to previously reported humidity sensors, the response and recovery time of this sensor are shorter. Figure 6c shows the dynamic response process from 0% RH to the remaining humidity. It can be found that the 5-mol%-Eu-TO sensor has good response and recovery ability at various humidities.

Long-term stability is a key indicator for measuring the reliability and reusability of a sensor. Figure 7 shows the long-term stability of the 5-mol%-Eu-TO humidity sensor at different RH values. The long-term stability test was completed within 28 days, with every 7 days representing a cycle. The impedance value fluctuated only slightly at 33% RH and 85% RH. Therefore, the 5-mol%-Eu-TO humidity sensor can be applied to various relative humidity environments and has the characteristic of long-term stability.

To study the adsorption behavior of a humidity sensor, it is important to study the response/recovery time and response of the sensor. Table 1 summarizes the humidity-sensing characteristics of different sensors that operate at room temperature within the last three years. Compared to other humidity sensors that have been reported, the Eu/TiO_2_ sensor in this work has a quick response/recovery time and high response when used for humidity detection at room temperature.

The humidity-sensing mechanism of the 5-mol%-Eu-TO humidity sensor can be further analyzed based on complex impedance spectroscopy (CIS) and the corresponding equivalent circuit (EC), as shown in Figure 8. During the tests, the drive voltage was set to be AC 1 V, and the frequency range was set to be 40~100 kHz. At 0% RH, the CIS shows a straight line in the high-frequency region, and the equivalent circuit can be composed of a constant solid-phase element (CPE), as shown in Figure 8a,d. In this environment, there are no adsorbed water molecules on the surface of 5 mol% Eu-TO, so that it is difficult to form a proton conduction process. Due to the inherent resistance, the 5-mol%-Eu-TO sensor exhibits a higher resistance value.

As shown in Figure 8b, at 33%, 43% and 59% RH, the CIS presents a semicircle, and a straight line gradually appears in the low-frequency region with the increase in RH. EC can be modeled by a parallel circuit consisting of resistance (R_f_) and capacitance (C_f_). The corresponding resistance component gradually decreases, and the capacitance component in Figure 8e gradually increases in EC. At this time, water molecules are adsorbed on hydroxyl groups by way of chemical and physical adsorption. The physically adsorbed water molecules cannot move freely under the action of hydrogen bonds, and a liquid water layer is gradually formed. With the adsorption of a lot of water molecules, ionized H^+^ and H_3_O^+^ are formed by jumping between the discontinuous water layers. At this stage, the transfer of protons becomes relatively easy, since the Grotthuss ion transport process (H_2_O + H_3_O^+^→H_3_O^+^ + H_2_O) [39,40] reduces the impedance.

At high humidity levels (75%, 85% and 95% RH), the semicircle of the CIS in Figure 8c gradually disappears in the high-frequency region, and it gradually appears as a trailing tail in the low-frequency region. This is caused by Warburg impedance generated by ion diffusion, and the corresponding equivalent circuit is a Warburg impedance (ZW) connected in series on the basis of the original circuit [41,42], as shown in Figure 8f. At this stage, a large amount of H_2_O is adsorbed via physical adsorption to form a continuous water film, and H_3_O^+^ ions are conducted by diffusion on the surface. According to the Grotthuss chain reaction mechanism, water molecules are ionized to form a large amount of H_3_O^+^. H_3_O^+^ continuously transfers H^+^ to adjacent water molecules to realize free migration of H^+^ on the surface of the material.

To explore the impact of Eu doping on the humidity response of TiO_2_, the structure changes from TiO_2_ to Eu/TiO_2_ illustrated in Figure 9 were calculated by density functional theory (DFT) and the Dmol3 procedure. In the DFT calculation of this work, the exchange-correlation functional was employed with revised Perdew–Burke–Ernzerhof (RPBE)-type generalized gradient approximation (GGA). Spin-unrestricted self-consistent field calculation was performed without symmetry restriction. Considering relativistic correction, DSPP (DFT Semi-core Pseudopots) was used to treat core electrons. During geometry optimization, convergence criterions were set to be 0.002 Hartree/Å (1 Hartree = 27.2 eV) for the maximum force, 0.005 Å for the maximum displacement and 10^−5^ Hartree for total energy. The SCF density convergence was set to be 5 × 10^−6^ Hartree with a thermal smearing of 0.005 Hartree. The TiO_2_ model contains 46/79 atoms (46 Ti atoms and 79 O atoms) and the Eu/TiO_2_ model contains 45/79 (45 Ti atoms and 79 O atoms).

Figure 9a shows the lattice plane (110) of rutile-type TiO_2_. Label 1 or 2 indicates the site where the Ti atom should be replaced by the Eu atom. Figure 9b shows the optimized structure when the Eu atom replaces the Ti atom labeled 1. Because the first, the second and the third ionization energy values of Eu (547.1, 1085 and 2404 KJ/mol) are much smaller than those of Ti (658.8, 1309.8 and 2652.5 KJ/mol), more O atoms are attracted to surround the Eu atom. Therefore, some Ti atoms lose their coordinate O atom, and Ti-Ti bonds are formed at the surface. Ever if Eu replaces the inner Ti atom labeled 2 as shown in Figure 9c, the movement of oxygen atoms to Eu causes similar structural changes at the surface. According to the Mulliken atomic charge, the charge of Eu in Figure 9c is about +1.68 |e| (|e| is the charge of a proton), and the spin of Eu is about −6.71 µ_B_ (µ_B_ is the Bohr magneton). Considering the outer shell configuration of Eu is 4f_7_/6s_2_, the calculated results indicate the Eu is a +2 ion with a half-filled 4f orbital. Compared to Figure 9a, the surface structure has been significantly altered in Figure 9b,c because the O atoms are close to Eu. The newly formed Ti-Ti bonds and broken O-Ti bonds indicate an O vacancy could be formed on the surface of TiO_2_, which means the introduction of Eu atoms can effectively increase the oxygen vacancy on the surface of TiO_2_.

Furthermore, the O atom surrounding the Eu atom in Figure 9b,c could attract a H^+^ ion in the solution to form a hydroxyl, which can help adsorb more water molecules because the hydroxyl group is a hydrophilic functional group. Moreover, the stable half-filled orbital of Eu may provide a weak capacity to adsorb water molecules, which further improves the water absorption of TiO_2_.

Therefore, the doped Eu atom can attract more oxygen atoms to produce oxygen vacancies and hydroxyls on the surface of TiO_2_, which can effectively improve the performance of the TiO_2_ humidity sensor. Moreover, the DFT explanation is more suitable for the role of proton conduction during the low-humidity state. Eu-doped TiO_2_ can produce more oxygen vacancies and hydroxyls, which will capture more water molecules and ionize more protons, thus increasing the sensor’s electrical conductivity and response.

## 4. Conclusions

In this work, a high-performance humidity sensor based on Eu-doped TiO_2_ was successfully prepared by hydrothermal synthesis, and the sensing mechanism was studied using DFT calculations. The experimental results show that doped Eu can cause more hydroxyls and oxygen vacancy defects on the surface of TiO_2_, thus enhancing the adsorption of water molecules and improving the humidity response of TiO_2_. In addition, the strong electric field around oxygen vacancies accelerates the dissociation of water molecules, so that more protons can migrate freely to improve the response speed of the sensor. When the doping mole ratio is 5 mol%, the response of the humidity sensor reaches to 23,997.0, the humidity hysteresis is 2.3% and the response/recovery time is 3/13.1 s. Furthermore, simulation results based on DFT indicate the doped Eu atom can produce more oxygen vacancies and hydroxyls on the surface of TiO_2_, which can effectively improve the performance of the TiO_2_ humidity sensor, which corresponds to the experimental results. This study not only fills the research gap on rare earth Eu-doped TiO_2_ as a humidity-sensitive material, but also proves that the presence of more oxygen vacancies and hydrophilic hydroxyl groups on the surface of metal oxides are conducive to the improvement of humidity performance.

## Figures and Tables

**Figure 1 sensors-24-04142-f001:**
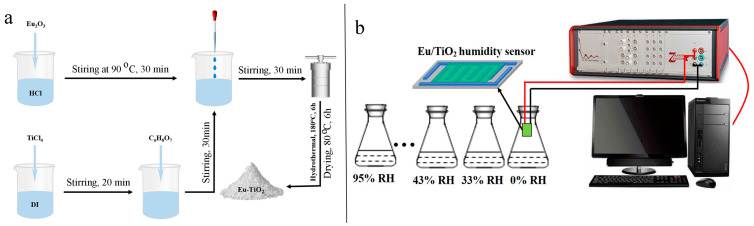
(**a**) The schematic flow diagram of the preparation of Eu-doped TiO_2_. (**b**) Schematic diagram of Eu/TiO_2_ humidity-sensing test system.

**Figure 2 sensors-24-04142-f002:**
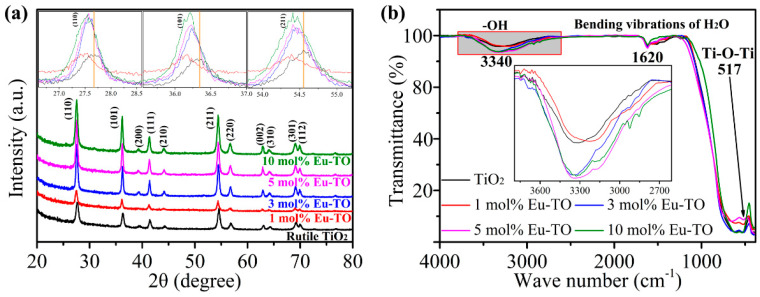
(**a**) XRD patterns of TiO_2_ and Eu-TiO_2_; (**b**) FTIR patterns of TiO_2_ and Eu-TiO_2_.

**Figure 3 sensors-24-04142-f003:**
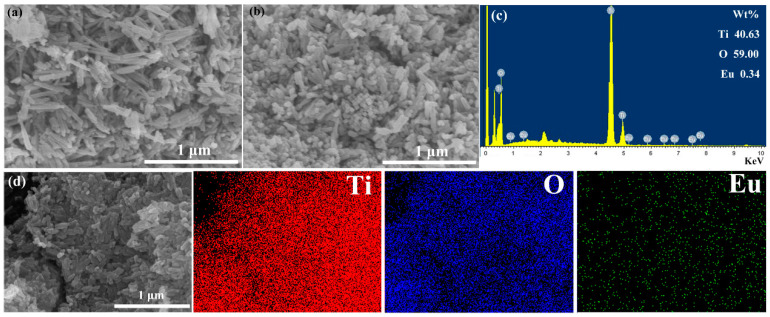
SEM images of (**a**) TiO_2_ and (**b**) 5-mol%-Eu-TO, (**c**) EDS analysis of 5-mol%-Eu-TO, (**d**) SEM-mapping image of 5-mol%-Eu-TO.

**Figure 4 sensors-24-04142-f004:**
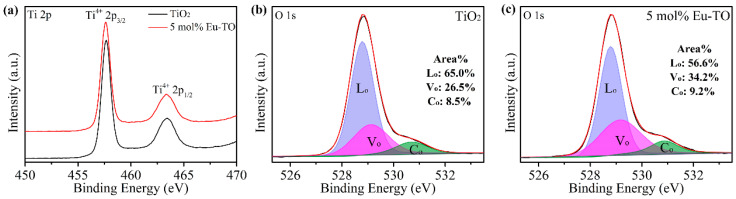
(**a**) Ti 2p spectra of TiO_2_ and 5-mol%-Eu-TO; O 1s spectra of (**b**) TiO_2_ and (**c**) 5-mol%-Eu-TO.

**Figure 5 sensors-24-04142-f005:**
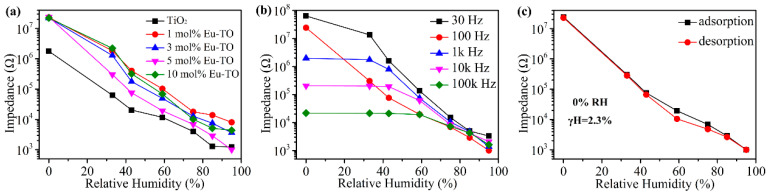
(**a**) Response curve of Eu-TiO_2_ humidity sensor, and (**b**) frequency selectivity and (**c**) hysteresis characteristic curve of 5-mol%-Eu-TO humidity sensor.

**Figure 6 sensors-24-04142-f006:**
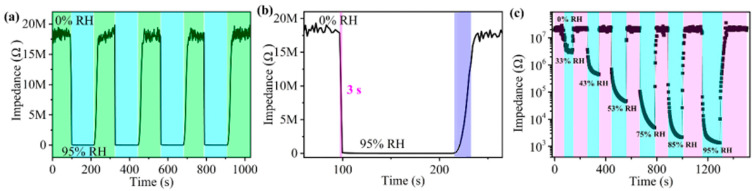
(**a**) Repeatability, (**b**) response and recovery time and (**c**) dynamic response curve of 5-mol%-Eu-TO humidity sensor.

**Figure 7 sensors-24-04142-f007:**
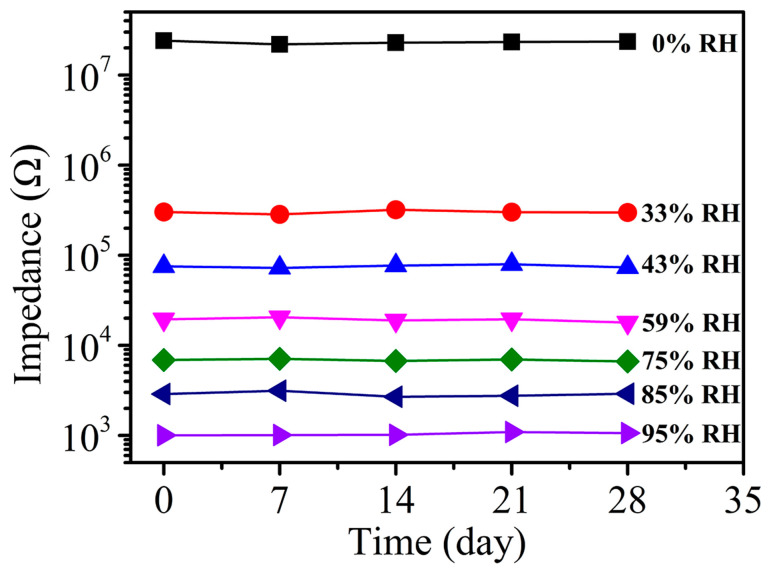
Long-term stability of 5-mol%-Eu-TO humidity sensor.

**Figure 8 sensors-24-04142-f008:**
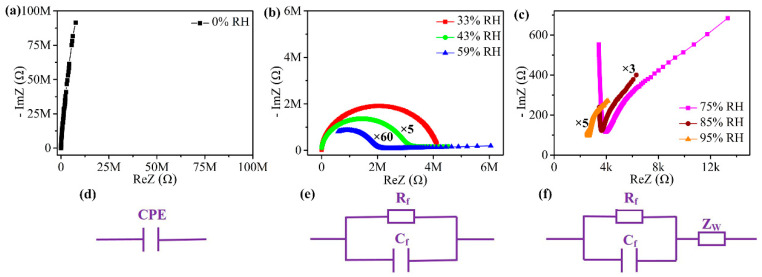
(**a**–**c**) Complex impedance spectroscopy, and (**d**–**f**) corresponding equivalent circuits of 5-mol%-Eu-TO humidity sensor at different RH values.

**Figure 9 sensors-24-04142-f009:**
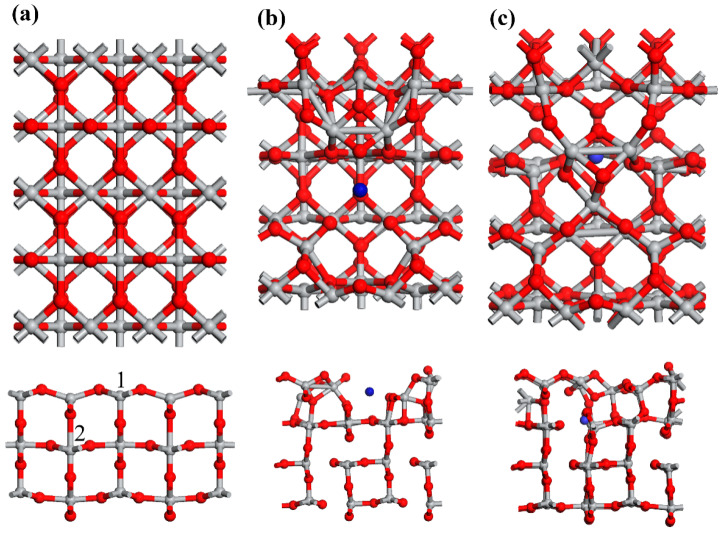
(**a**) Lattice plane (110) of pristine TiO_2_. (**b**) or (**c**) Optimized structure of Eu-doped TiO_2_ when a Ti atom labeled by 1 or 2 in (**a**) is replaced by a Eu atom. Red, gray and blue balls represent O, Ti and Eu atoms, respectively.

**Table 1 sensors-24-04142-t001:** Comparison of performance of humidity sensors reported within the last three years.

Materials	Response	Res./Rec. Time (s)	Sensing Range (% RH)	Refs.
rGO/Fe_2_O_3_	2715.541%	5 s/10 s	40–80% RH	[7]
ZrP	91.5%	57 s/9 s	10.9–91.5% RH	[31]
PANI/RGO	10.52%	3 s/4 s	7–97% RH	[32]
CS/GO/SnO_2_	72,683%	8 s/8 s	15–95% RH	[33]
TiO_2_/CdS	63,459	5 s/7 s	11–95% RH	[34]
Ag/SnO_2_	24,292.3	5/8 s	11–95% RH	[35]
PDDA/Ti_3_C_2_T_x_	48,813%	8.794 s/2.656 s	11–97% RH	[36]
Fe/SnO_2_	145,290	10 s/8 s	11–95% RH	[5]
SnO_2_/rGO	3.84 × 10^5^%	45 s/8 s	11–95% RH	[37]
KCl/Sm_2_O_3_	127,121	0.1 s/21 s	11–95% RH	[38]
Eu/TiO_2_	23,997	3 s/13.1 s	11–95% RH	This work

## Data Availability

Data available in a publicly accessible repository.

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
