# Peer review of "Preparation and Mechanism Analysis of High-Performance Humidity Sensor Based on Eu-Doped TiO2"

_sensors, 2024, doi:10.3390/s24134142_

Round 1
Reviewer 1 Report
Comments and Suggestions for Authors
In this manuscript (sensors-3057875), the authors reported a humidity sensor based on Eu-doped TiO2. The humidity sensing performance are generally acceptable, but there are many issues in writing, organization structure, results and discussion. The manuscript may be accepted after addressing the following issues.
1. Abstract, line 55 and results: Response and sensitivity cannot be confused. Sensitivity is the slope of the linear fitting equation between response and RH (). Suggest using response (order of magnitude of change) instead of sensitivity.
2. Line 12: “undoped TiO2”. The numbers in the chemical formula require subscripts. Check the entire text, including references.
3. Introduction: (1) “…excellent surface volume ratio, high surface activity, low production cost, easy measurement and so on”. Some of the viewpoints are not correct. There is no advantage in specific surface area, for example, it cannot compare to two-dimensional materials. (2) What is the problem addressed in this work regarding humidity sensors? In fact, compared to oxides, many humidity sensing materials and humidity sensors currently exhibit excellent humidity sensing performances, which the authors have not discussed (such as halloysite nanotubes, attapulgite, sepiolite nanofibers, zirconium phosphate nanoplates, and MXenes). (3) There are many types of humidity sensors (such as resistance, capacitance, impedance, and voltage (electrochemical, ion gradient)), why fabricate and investigate impedance humidity sensors?
4. Fig. 2: Suggest placing it in the mechanism analysis section after Fig. 9.
5. Fig. 6 (a): It needs to provide the operating frequency.
6. The selective test in Fig. 8 (b) is meaningless and it is recommended to delete it. If expressed in ppm, humidity (thousands to tens of thousands of ppm) is much higher than 100 ppm.
7. To confirm the high performance in the title, it is necessary to compare it with more than 10 references of humidity sensors from the past three years, using a table.
8. Fig. 9: It is necessary to combine sensing performance (Fig.6 (b)) and complex impedance plots for sensing mechanism analysis, refering to Sens. Actuators, B Chem., 2023, 394, 134445. In addition, supporting references are needed.
9. The reference format does not meet the requirements of the journal. Authors and journal names need to be abbreviated. In addition, most of the references are outdated, and it is recommended to focus on references from the past three years.
10. Please improve English writing and check the format of the journal.
Comments on the Quality of English LanguageMinor editing of English language required.
Author Response
We appreciate very much the careful reading of our manuscript and valuable suggestions of you. We have carefully considered the comments and have revised the manuscript accordingly. We have revised the paper according to your comments. The followings are the answers and revisions we have made in response to your questions and suggestions on an item by item basis.
- Abstract, line 55 and results: Response and sensitivity cannot be confused. Sensitivity is the slope of the linear fitting equation between response and RH (). Suggest using response (order of magnitude of change) instead of sensitivity.
Response: Thanks for your suggestion. According to the definitions of sensitivity and response given, sensitivity is replaced by response in the paper. This part can be found in lines 14, 78, 351 and 355.
- Line 12: “undoped TiO2”. The numbers in the chemical formula require subscripts. Check the entire text, including references.
Response: I am sorry for this mistake. In the revised paper, we have re-examined the writing and formatting of the full text and corrected it.
- Introduction: (1) “…excellent surface volume ratio, high surface activity, low production cost, easy measurement and so on”.Some of the viewpoints are not correct. There is no advantage in specific surface area, for example, it cannot compare to two-dimensional materials. (2) What is the problem addressed in this work regarding humidity sensors? In fact, compared to oxides, many humidity sensing materials and humidity sensors currently exhibit excellent humidity sensing performances, which the authors have not discussed (such as halloysite nanotubes, attapulgite, sepiolite nanofibers, zirconium phosphate nanoplates, and MXenes). (3) There are many types of humidity sensors (such as resistance, capacitance, impedance, and voltage (electrochemical, ion gradient)), why fabricate and investigate impedance humidity sensors?
Response: Thank you for your suggestion. In the revised paper, we rewritten the introduction. There are many types of humidity sensors, such as resistance, capacitance, impedance, and voltage (electrochemical, ion gradient), Among these sensor types, impedance humidity sensor has high precision, good heat dissipation and large nonlinearity, so they can be widely used in automatic test and control technology. For humidity sensor, sensitive material directly interacts with target molecules and has an important influence on performance. Compared to metal oxide semiconductor (MOS), many humidity sensing materials, such as halloysite nanotubes, attapulgite, sepiolite nanofibers, zirconium phosphate nanoplates, and MXenes also exhibit excellent humidity sensing performances. But in fact, MOS has been widely used in humidity sensitive material due to its advantages of schottky contact can generate more charge transfer between water molecules and MOS, and MOS is usually composed of particles with grain boundary barriers between particles. This part can be found in line 24-36.
- 2: Suggest placing it in the mechanism analysis section after Fig. 9.
Response: Thank you for your suggestion. In the revised paper, the calculation and mechanism analysis section was moved to the end of the discussion and updated the figure sequence number.
- 6 (a):It needs to provide the operating frequency.
Response: Thank you for your suggestion. In the revised paper, the operating frequency is 100 Hz was added in line 202.
- The selective test in Fig. 8 (b) is meaningless and it is recommended to delete it. If expressed in ppm, humidity (thousands to tens of thousands of ppm) is much higher than 100 ppm.
Response: Thank you for your suggestion. In the revised paper, Fig. 8(b) has been removed.
- To confirm the high performance in the title, it is necessary to compare it with more than 10 references of humidity sensors from the past three years, using a table.
Response: Thank you for your suggestion. In the revised paper, 10 references of humidity sensors from the past three years in table 1 was added to the revised paper. Table 1 summarizes the humidity sensing characteristics of different sensors within the last three years. Compared to other humidity sensors that have been reported, Eu/TiO2 sensor in this work has a quick response/recovery time and high response when used for humidity detection at room temperature.This part can be found in line 274-279.
- 9: It is necessary to combine sensing performance (Fig.6 (b)) and complex impedance plots for sensing mechanism analysis, refering to Sens. Actuators, B Chem., 2023, 394, 134445. In addition, supporting references are needed.
Response: Thank you for your suggestion. In the revised paper, Fig. 5(b) shows that higher frequencies reduce the sensor’s performance, which is mainly because the polarization direction of water molecules is difficult to keep consistent with change direction of high-frequency electric field. Meanwhile, the polarization speed of water molecules is difficult to keep up with changes of electrical field direction at high working frequency, resulting in a decrease of the dielectric constant and a decrease in its dependence on RH. This part can be found in line 229-235 and the refer article has been added to Refs [30].
- The reference format does not meet the requirements of the journal. Authors and journal names need to be abbreviated.In addition, most of the references are outdated, and it is recommended to focus on references from the past three years.
Response: Thank you for your suggestion. In the revised paper, we have revised the format of all references.
- Please improve English writing and check the format of the journal.
Response: In the revised draft, we have re-checked the English and format, hoping to meet your requirements.
Reviewer 2 Report
Comments and Suggestions for Authors
The paper on preparation and mechanism analysis of high performance humidity sensor based on Eu-doped TiO2 presents both simulation nd experimental to investigate the performance humidity sensor based on Eu-doped TiO2 . This manuscript still has very serious problems relating to novelty. The key issues are 1) the authors did not demonstrate/show the detail calculation using density functional theory (DFT) study to investigate the feasibility of TiO2 doped by Eu atom for computing. Only tA Dmol3 package based on DFT was used to investigate the changes in geometry 97 structure are not enough to understand the general readership of the paper.
Moreover, some description such as section 2.2 Preparation of Eu-Tio2 needs more clarification with schematic flow diagram. The description of section 2.3 Humidity sensor tests is also not enough.
To study the adsorption behavior, response and recovery speed of sensor, it is important to provide references.
Working frequency is also an important indicator to measure the performance of a sensor. Please explain it with reference.
The description of results of measurements and computations, discussion should be more in details if this paper was submitted for a regular issue without page limit. Specially figures need appropriate discussion even though I strongly recommend the revision of manuscript. A notation list is necessary and equations numbers are also missing.
In conclusion, the calculations indicates that the proposed material can be promising for humidity sensors. If it is possible to provide the requirements for candidates for sensors in the introduction and can be related to the findings in results section with the calculation, then the article will be more attractive to the potential readers.
Comments on the Quality of English Language
There are some typos such as TiO2, H2O+H3O+→H3O++H2O etc.
Line 139: All materials exhibit 11 characteristic diffraction peaks of rutile 139 phase TiO2 (JCPDS card No. 21-1276. But there is no 11 characteristic diffraction peaks.
Author Response
We appreciate very much the careful reading of our manuscript and valuable suggestions of you. We have carefully considered the comments and have revised the manuscript accordingly. We have revised the paper according to your comments. The followings are the answers and revisions we have made in response to your questions and suggestions on an item by item basis.
The paper on preparation and mechanism analysis of high performance humidity sensor based on Eu-doped TiO2 presents both simulation nd experimental to investigate the performance humidity sensor based on Eu-doped TiO2. This manuscript still has very serious problems relating to novelty. The key issues are
1)the authors did not demonstrate/show the detail calculation using density functional theory (DFT) study to investigate the feasibility of TiO2 doped by Eu atom for computing. Only tA Dmol3 package based on DFT was used to investigate the changes in geometry 97 structure are not enough to understand the general readership of the paper.
Response: Thank you for your suggestion. We have reorganized the calculation and added the reasons for the gas sensitive properties of europium-doped titanium oxide. Specifically as follows:
To explore the impact of Eu doping on the humidity sensitivity of TiO2, the structure changes from TiO2 to Eu/TiO2 are illustrated in Fig. 9 was calculated by density-functional theory (DFT) and the Dmol3 procedure. In the DFT calculation of this work, exchange-correlation functional was employed with revised Perdew-Burke-Ernzerhof (RPBE)-type generalized gradient approximation (GGA). Spin-unrestricted self-consistent field calculation was done without symmetry restriction. Considered relativistic correction, DSPP (DFT Semi-core Pseudopots) was used to treat core electrons. During geometry optimization, convergence criterions were set to be 0.002 Hartree/Å (1 Hatree=27.2 eV) for the maximum force, 0.005 Å for the maximum displacement and 10-5 Hartree for total energy. SCF density convergence was set to be 5×10-6 Hartree with a thermal smearing of 0.005 Hatree. TiO2 model contains 46/79 atoms (46 Ti atoms and 79 O atoms) and Eu/TiO2 model contains 45 /79 (45 Ti atoms and 79 O atoms).
Fig. 9(a) shows lattice plane (110) of rutile type TiO2. Label 1 or 2 indicates the site where Ti atom should be replaced by Eu atom. Fig. 9(b) shows optimized structure when Eu atom replaces Ti atom labelled 1. Because the first, the second, and the third ionization energy of Eu (547.1, 1085, 2404 KJ/mol) are much smaller than that of Ti (658.8, 1309.8 and 2652.5 KJ/mol), more O atoms are attracted to surround Eu atom. Therefore, some Ti atoms lose their coordinate O atom and Ti-Ti bonds are formed at the surface. Ever if Eu replaces inner Ti atom labelled 2 as shown in Fig. 9(c), the move of oxygen atoms to Eu causes similar structural changes at the surface. According to Mulliken atomic charge, charges of Eu in Fig. 9(c) is about +1.68 |e| (|e| is the charge of a proton), and spin of Eu is about -6.71 µB (µB is Bohr magneton). Considering outer shell configuration of Eu is 4f7/6s2, the calculated results indicate the Eu is a +2 ion with a half-filled 4f orbital. Comparing to Fig. 9(a), surface structure has been significantly altered in Fig. 9(b) and (c) because O atoms is close to Eu. The new formed Ti-Ti bonds and broken O-Ti bonds indicate O vacancy could be formed on the surface of TiO2, which means the introduction of Eu atoms can effectively increases the oxygen vacancy on the surface of TiO2.
Furthermore, the O atom surround Eu atom in Fig. 9(b) and (c) could attract H+ ion in solution to form a hydroxyl, which can help adsorb more water molecules result in hydroxyl is a hydrophilic functional group. Moreover, the stable half-filled orbital of Eu may provide a weak capacity to adsorb water molecules, which further improves the water absorption of TiO2.
Therefore, the doped Eu atom could attract more oxygen atom to produce oxygen vacancies and hydroxyls on the surface of TiO2, which can effectively improve the performance of TiO2 humidity sensor. Moreover, DFT explanation more suitable for the role of proton conduction during the low humidity state. Eu doped TiO2 can produce more oxygen vacancies and hydroxyls, which will capture more water molecules and ionize more protons, thus increase the sensor's electrical conductivity and sensitivity. This part can be found in line 313-352.
2)Moreover, some description such as section 2.2 Preparation of Eu-Tio2 needs more clarification with schematic flow diagram. The description of section 2.3 Humidity sensor tests is also not enough.
Response: Thank you for your suggestion. The schematic flow diagram of preparation of Eu-doped TiO2 has been added to revised paper. This part can be found in Fig. 1(a). Furthermore, a detailed description of the humidity sensor tests has also been added to the revised manuscript. This part can be found in line 114-128.
- To study the adsorption behavior, response and recovery speed of sensor, it is important to provide references.
Response: Thank you for your suggestion. To study the adsorption behavior of humidity sensor, it is important to study the response /recovery time and sensitivity of sensor. 10 references has been added in Table 1in revised paper. Table 1 summarizes the humidity sensing characteristics of different sensors within the last three years. Compared to other humidity sensors that have been reported, Eu/TiO2 sensor in this work has a quick response/recovery time and high response when used for humidity detection at room temperature. This part can be found in line 268-275.
- Working frequency is also an important indicator to measure the performance of a sensor. Please explain it with reference.
Response: Thank you for your suggestion. Fig. 5(a) shows response of TiO2 and Eu-TiO2 humidity sensors in the range of 0% ~95% RH, and the operating frequency is 100 Hz. Fig. 5(b) shows that higher frequencies reduce the sensor’s performance, which is mainly because the polarization direction of water molecules is difficult to keep consistent with change direction of high-frequency electric field. Meanwhile, the polarization speed of water molecules is difficult to keep up with changes of electrical field direction at high working frequency, resulting in a decrease of the dielectric constant and a decrease in its dependence on RH [30]. This part and the corresponding references [30] have been added to the revised draft, which can be found in line 223-229.
- The description of results of measurements and computations, discussion should be more in details if this paper was submitted for a regular issue without page limit. Specially figures need appropriate discussion even though I strongly recommend the revision of manuscript. A notation list is necessary and equations numbers are also missing.
Response: Thank you for your suggestion. We have re-written the introduction and calculation part, supplemented the test details of the humidity sensor, the influence of different frequencies on the performance of the sensor, and the performance of the metal oxide humidity sensor in the past two years, hoping that the revised draft can meet the requirements of the journal.
- In conclusion, the calculations indicates that the proposed material can be promising for humidity sensors. If it is possible to provide the requirements for candidates for sensors in the introduction and can be related to the findings in results section with the calculation, then the article will be more attractive to the potential readers.
Response: Thank you for your suggestion. We have rewritten the conclusions and summarized the characteristics of the calculated results for high performance metal oxide-based humidity-sensitive materials, hoping to be helpful to interested readers. Conclusions are as follows:
In this work, a high performance humidity sensor based on Eu-doped TiO2 are successfully prepared by hydrothermal and the sensing mechanism was studied by DFT calculation. Experimental results show that doped Eu can cause more hydroxyls and oxygen vacancy defects on surface of TiO2, thus enhancing adsorption of water molecules and improving humidity response of TiO2. In addition, the strong electric field around oxygen vacancy accelerate dissociation of water molecules, so that more protons can migrate freely to improve response speed of the sensor. When doping mole ratio is 5 mol%, response of the humidity sensor reaches to 23997.0, humidity hysteresis is 2.3%, and response/recovery time is 3/11 s. Furthermore, simulation results based on DFT indicate the doped Eu atom can produce more oxygen vacancies and hydroxyls on the surface of TiO2, which can effectively improve the performance of TiO2 humidity sensor, which corresponds to the experimental results. This study not only supplements the blank of rare earth Eu doped TiO2 as humidity sensitive material, but also prove that more oxygen vacancies and hydrophilic hydroxyl groups on the surface of metal oxides are conducive to the improvement of humidity performance. This part can be found in line 359-373.
Reviewer 3 Report
Comments and Suggestions for Authors
This paper demonstrates that Eu-doped TiO2 nanorods, synthesized through hydrothermal synthesis, exhibit enhanced humidity sensitivity, faster response speed, long-term stability, and good anti-interference ability for resistive humidity sensors due to increased oxygen vacancies and hydroxyl defects, making them a promising humidity-sensitive material.
1- Include a brief statement on why Eu-doping specifically is chosen over other RE elements and clearly articulate the research gap.
2- Expand the literature review to include recent studies on Eu-doped TiO2.
3- Elaborate on the rationale behind choosing DFT for theoretical studies and hydrothermal synthesis for experimental preparation at the end of the introduction.
4- Refer to the articles that investigate the effect of humidity on the sensing performance of TiO2 (DOI:10.1088/1361-6528/abfd54).
5- The sources of the materials are mentioned, but the purity of the chemicals is not specified.
6- Mention details in some steps of the synthesis procedure (e.g., the exact cooling time, centrifugation speed, and duration).
7- Mention equilibration time at each humidity level.
8- The quantitative change in the number of oxygen vacancies with different Eu concentrations is not provided.
9- Why do higher frequencies reduce the sensor's performance?
10- Can you describe the roles of proton conduction, water layer formation, and the Grotthuss mechanism in your theoretical model?
Comments on the Quality of English LanguageSome sentences are complex and can be simplified for better readability. For example, a sentence starting with " As one of the most active RE metal elements..." is too long.
Author Response
We appreciate very much the careful reading of our manuscript and valuable suggestions of you. We have carefully considered the comments and have revised the manuscript accordingly. We have revised the paper according to your comments. The followings are the answers and revisions we have made in response to your questions and suggestions on an item by item basis.
- Include a brief statement on why Eu-doping specifically is chosen over other RE elements and clearly articulate the research gap.
Response: Eu doping makes it easier to modulate semiconductor bandgap width. Furthermore, Eu ions doping may contribute to locate some Eu rich composition on the surface of TiO2 and further increase surface oxygen vacancies and hydrophilic groups of TiO2, which can absorb more water molecules to enhance the sensitivity of TiO2 humidity sensor. This part can be found in line 55-59.
- Expand the literature review to include recent studies on Eu-doped TiO2.
Response: Thank you for your suggestion. In the revised paper, some latest literatures [20-22] have been added to the introduction in the revised paper. This part can be found in line 55-59.
- Elaborate on the rationale behind choosing DFT for theoretical studies and hydrothermal synthesis for experimental preparation at the end of the introduction.
Response: Thank you for your suggestion. In the revised paper, DFT detail for theoretical studies and hydrothermal synthesis were added in the introduction. The calculation shows that the introduction of Eu ions into TiO2 can form more oxygen vacancies on the surface of TiO2. Furthermore, the oxygen atoms around Eu atoms can attract hydrogen ions in the environment to form hydroxyl, which is conducive to the capture of water molecules. In experiments, Eu-doped TiO2 resistive humidity sensors with different mole ratios were prepared by hydrothermal synthesis to achieve better humidity sensing performance. Hydrothermal synthesis is relatively simple, does not need high temperature treatment can obtain a complete crystal shape, uniform particle size, good dispersiont, thus relatively reduce energy consumption. In addition, the required raw materials are relatively cheap and easy to obtain, and the resulting products have uniform phase, high purity, good crystallization, high yield, and controllable product morphology and size. This part can be found in line 67-77.
- Refer to the articles that investigate the effect of humidity on the sensing performance of TiO2(DOI:10.1088/1361-6528/abfd54).
Response: Thank you for your suggestion. The increase in the conductance of TiO2 by increasing the RH can be described as a sequence of dissociation and reduction. The reaction constitutes the homolytic dissociation of H2O and the reaction of the neutral H atom with the lattice oxygen. In addition to ionizing the lattice, the reaction releases the electron as a donor to the rooted OH group. This part can be found in line 42-46 and the refer article has been added to Refs [13].
- The sources of the materials are mentioned, but the purity of the chemicals is not specified.
Response: Thank you for your suggestion. The reagents in the experiment were all analytical grade (AR) and did not require further purification for use. This part can be found in line 87-88.
- Mention details in some steps of the synthesis procedure (e.g., the exact cooling time, centrifugation speed, and duration).
Response: Thank you for your suggestion. The mixture was cooled down to room temperature for 2 h, before the precipitate was centrifuged at 8000 rpm with DI and anhydrous ethanol respectively for 10 min and repeated three times. This part can be found in line 105-107.
- Mention equilibration time at each humidity level.
Response: Thank you for your suggestion. The sensor's equilibrium time at each humidity level is approximately 100 s. This part can be found in line 121-123.
- The quantitative change in the number of oxygen vacancies with different Eu concentrations is not provided.
Response: Thank you for your suggestion. But we are sorry that it is difficult for us to complete the XPS test of all samples within one week. The appointment time for XPS is very long, so we cannot complete it in a short time. According to the properties of the humidity test in Fig. 5(a), the samples with the best humidity performance and pure TiO2 were selected for XPS testing. It can be shown that the surface oxygen vacancy of europium-doped zinc oxide is increased, and the high response to water molecules is also consistent with the calculated results. If we must provide XPS testing, please give us about 40 days.
- Why do higher frequencies reduce the sensor's performance?
Response: Thank you for your question. Fig. 5(b) shows that higher frequencies reduce the sensor’s performance, which is mainly because the polarization direction of water molecules is difficult to keep consistent with change direction of high-frequency electric field. Meanwhile, the polarization speed of water molecules is difficult to keep up with changes of electrical field direction at high working frequency, resulting in a decrease of the dielectric constant and a decrease in its dependence on RH. This part can be found in line 229-235
- Can you describe the roles of proton conduction, water layer formation, and the Grotthuss mechanism in your theoretical model?
Response: Thank you for your suggestion. DFT explanation applies to the role of proton conduction during the low humidity state. Eu doped TiO2 can produce more oxygen vacancies, which will capture more water molecules and ionize more protons, thus increase the sensor's electrical conductivity and improve the sensor’s sensitivity. This part can be found in line 337-340.
Round 2
Reviewer 1 Report
Comments and Suggestions for Authors
The following issues still exist:
1. Title: Except for the first word in the title, the first letter of other words also need to be capitalized (Except for prepositions and articles).
2. Items related to “sensitivity” need to be deleted. Check the entire text.
3. “There are many types…also exhibit excellent humidity sensing performances.” Supporting references are required.
4. “But in fact, MOS has been widely used in humidity sensitive material due to its advantages of schottky contact can generate more charge transfer between water molecules and MOS, and MOS is usually composed of particles with grain boundary barriers between particles.” What is the basis?
5. Figure 5b: It is unreasonable that there is a crossover in the responses at different frequencies. The frequency and impedance values are usually monotonic.
6. The definition of humidity hysteresis does not conform to the mainstream representation in the field of humidity sensors. The unit of humidity hysteresis is generally RH. Figure 5c shows that the humidity hysteresis is greater than 5% RH at 75% RH. Otherwise, the humidity hysteresis in Table 1 is not standardized uniformly.
7. Figure 6b: If the response recovery time is defined as T90, it is obvious that the recovery time is greater than 11s (the starting position of the shaded area is incorrect).
8. Table 1: All comparative references are at room temperature, so emphasizing room temperature is meaningless. Carefully check if the reference parameters in the table are correct.
9. Check the format of the references, such as abbreviating the journal name. All author names need to be provided.
Comments on the Quality of English LanguageMinor editing of English language required
Author Response
Dear reviewer
Re: Preparation and Mechanism Analysis of High Performance Humidity Sensor based on Eu-doped TiO2
Manuscript ID: sensors-3057875
Thank you very much for your letter and the comments on our paper submitted to Sensors.
We have learned much more from your comments, which are fair, encouraging and constructive.
After carefully studying your advice, we have made corresponding changes to the paper. I hope this will make it more acceptable for publication.
Our response of the comments is enclosed at the end of this letter.
If you have any questions about this paper, please don’t hesitate to contact us via email: zhanghyxj@163.com, Tel.: 13999839908.
We appreciate very much the careful reading of our manuscript and valuable suggestions of you. We have carefully considered the comments and have revised the manuscript accordingly. We have revised the paper according to your comments. The followings are the answers and revisions we have made in response to your questions and suggestions on an item by item basis.
- Title: Except for the first word in the title, the first letter of other words also need to be capitalized (Except for prepositions and articles).
Response: Thank you for your suggestion. The title has been revised as required in the revised paper.
- Items related to “sensitivity” need to be deleted. Check the entire text.
Response:Thank you for your suggestion. We have changed the sensitivity to response in the revised paper.
- “There are many types…also exhibit excellent humidity sensing performances.”Supporting references are required.
Response: Thank you for your suggestion. Supporting references are added in the revised paper. This part can be found in reference [2,3,5,6].
- “But in fact, MOS has been widely used in humidity sensitive material due to its advantages of schottky contact can generate more charge transfer between water molecules and MOS, and MOS is usually composed of particles with grain boundary barriers between particles.” What is the basis?
Response: Thank you for your suggestion. This sentence is in the learning process, my personal understanding, is indeed not suitable for the introduction, so we re-wrote the reasons for choosing MOS materials. But in fact, MOS has been widely used in humidity sensitive material due to its adjustable band gap, large specific surface area, abundant surface defect and reasonable charge transfer ability [5]. Furthermore, MOS-based humidity sensors shows notable features like as small size, easy synthesis and low cost compared with other types of humidity sensors [7]. This part can be found in line 32-38.
- Figure 5b: It is unreasonable that there is a crossover in the responses at different frequencies. The frequency and impedance values are usually monotonic.
Response: Thank you for your suggestion. First, the sensor's response to humidity at different frequencies is the actual measurement. Secondly,the frequency and impedance values of sensor at different working frequency are usually monotonic in Fig.5. Finally, it is possible for a small number of intersections to appear between the impedance changes tested at 100 Hz and those at other frequencies, because the impedance values tested at different frequencies may be the same. For example, a similar diagram appears in literatures [5, 34, 35, 37], but these intersections do not indicate that the frequency and impedance changes are not monotony. This part can be found in line 221-222 in the revised paper.
- The definition of humidity hysteresis does not conform to the mainstream representation in the field of humidity sensors. The unit of humidity hysteresis is generally RH. Figure 5c shows that the humidity hysteresis is greater than 5% RH at 75% RH. Otherwise, the humidity hysteresis in Table 1 is not standardized uniformly.
Response: Thank you for your suggestion. The method of calculating humidity hysteresis selected in this paper is only the same as reference 5, 34, 35, 37, and indeed cannot be compared with all references in the table. Therefore, we have removed the comparison of humidity hysteresis loops, and only emphasized that europium doping can greatly improve the humidity hysteresis of titanium dioxide in the revised paper. This part can be found in table 1.
- Figure 6b: If the response recovery time is defined as T90, it is obvious that the recovery time is greater than 11s (the starting position of the shaded area is incorrect).
Response: Thank you for your suggestion. We corrected the recovery time in Figure 6 and the recovery time is 13.1s in the revised paper. This part can be found in line 256 and line 289.
- Table 1: All comparative references are at room temperature, so emphasizing room temperature is meaningless.Carefully check if the reference parameters in the table are correct.
Response: Thank you for your suggestion. In Table 1, the comparative literature of the selected humidity sensor was all measured at room temperature, so we removed the temperature in the revised draft and only emphasized the test temperature in the description. This part can be found in line 292.
At the same time, we did a careful review of all the comparative humidity sensor in literature and added the test range of each humidity sensor.
- Check the format of the references, such as abbreviating the journal name.All author names need to be provided.
Response: Thank you for your suggestion. The reference format has been rechecked as requested and the names of all authors have been added.

Reviewer 2 Report
Comments and Suggestions for Authors
The authors addressed most of my comments and manuscript is improved.
Author Response
Dear reviewer
Re: Preparation and Mechanism Analysis of High Performance Humidity Sensor based on Eu-doped TiO2
Manuscript ID: sensors-3057875
Thank you very much for your letter and the comments on our paper submitted to Sensors.
We have learned much more from your comments, which are fair, encouraging and constructive.
After carefully studying your advice, we have made corresponding changes to the paper. I hope this will make it more acceptable for publication.
Our response of the comments is enclosed at the end of this letter.
If you have any questions about this paper, please don’t hesitate to contact us via email: zhanghyxj@163.com, Tel.: 13999839908.
We appreciate very much the careful reading of our manuscript and valuable suggestions of you. We have carefully considered the comments and have revised the manuscript accordingly. We have revised the paper according to your comments. The followings are the answers and revisions we have made in response to your questions and suggestions on an item by item basis.
Suggestions for Authors
The authors addressed most of my comments and manuscript is improved.
Response: Thank you for your suggestion.